# Diversity of Chemical Composition and Morphological Traits of Eight Iranian Wild *Salvia* Species during the First Step of Domestication

Ghasem Esmaeili [1] , Hamideh Fatemi [1] , Mahnaz Baghani avval [1], Majid Azizi [1,*] , Hossein Arouiee [1], Jamil Vaezi [2] and Yoshiharu Fujii [3,*]

1 Department of Horticultural Science, Faculty of Agriculture, Ferdowsi University of Mashhad, Mashhad 9177948974, Iran
2 Department of Biology, Faculty of Science, Ferdowsi University of Mashhad, Mashhad 9177948974, Iran
3 International Environmental and Agricultural Sciences, Fuchu Campus, Tokyo University of Agriculture and Technology, Tokyo 183-0054, Japan
* Correspondence: azizi@um.ac.ir (M.A.); yfujii@cc.tuat.ac.jp (Y.F.)

**Abstract:** As one of the largest genera of the Lamiaceae family, *Salvia* has a wide distribution worldwide. Despite their great importance and medicinal use, most *Salvia* species are collected from their natural habitats, and some of them are endangered and vulnerable. This study aimed to evaluate the domestication process of eight Iranian native *Salvia* species. The studied species were cultivated and adapted to the cultivation area after two years, and then some of their important biochemical properties were investigated. According to some significant results, the root architecture was closely correlated with the climatic conditions of the species origins. The distribution of total dry matter varied widely among species; accordingly, *S. sclarea* and *S. officinalis* had 65.6% and 55.9% dry weights in their leaves, respectively. Moreover, *S. nemorosa* had a 24.3% dry weight in its flowers, while *S. frigida* (Jahrom), *S. frigida* (Targavar), *S. virgata* (Eghled), and *S. macrosiphon* had 44.6%, 43.3%, 46.0%, and 44.3% dry weights in their roots. The most potent antioxidant activity (IC50) was observed in the roots of *S. macrosiphon* (10.9 μg/mL) and *S. sclarea* (14.9 μg/mL), the stem of *S. nemorosa* (14.3 μg/mL), and the leaves of *S. atropatana* (14.0 μg/mL). Rosmarinic acid, a key phenolic substance in *Salvia* species, was present in the range of 0.24–0.47 mg/g dry weight. The essential oil content ranged from 0.35% in *S. atropatana* to 1.45% (*w*/*w*) in *S. officinalis*. β-caryophyllene, caryophyllene oxide, and germacrene D were the major ingredients of the essential oils. The cluster analysis based on the essential oil data revealed the most similarities between *S. sclarea* and *S. macrosiphon*, and a clear separation of *S. virgate*, *S. syriaca*, and *S. officinalis* from other species. *Salvia* spp. contain a wide variety of compounds of interest under cultivation, with *S. sclarea* having the greatest potential to profit from the production of medicinal compounds, such as phenolic compounds, flavonoids, and essential oils. Furthermore, *S. officinalis*, *S. nemorosa*, and *S. sclarea* are the best species for producing raw medicinal materials.

**Keywords:** essential oil; inter-specific variation; root architecture; *Salvia* spp.

## 1. Introduction

Medicinal and aromatic plants and mushrooms are considered the most important and largest sources of secondary metabolites [1–3]. According to reports of the World Health Organization, up to 80% of people all over the world use medicinal plants [4]. Today, there is a great deal of interest in the study, discovery, and use of the biological effects of the useful and valuable metabolites of medicinal plants [5]. Although in most European countries *Salvia* is cultivated in the field, some medicinal plants are being harvested from nature, not only for traditional medicinal purposes but also for trade and commerce [2,6,7]. According to reports of the International Union for Conservation of Nature, about 15,000 plant

species are threatened with extinction due to intensive harvesting and destruction of their natural habitat [8]. Therefore, the cultivation of medicinal plants is needed to meet high demand and conserve genetic diversity. Cultivation would also reduce the variation and contamination of plant materials [9].

With more than 1000 species worldwide, *Salvia* is the largest genus of the *Lamiaceae* family [10]. *Salvia* species with herbaceous, annual, biennial, perennial, and shrub growth forms are distributed in various ecological areas. Due to their diverse and widespread nature, *Salvia* species are a significant and various source of secondary metabolites, such as flavonoids and terpenoids [11,12].

These plants have many uses in folk medicine, perfumery, and the food and cosmetic industries [13,14]. Various biological effects of *Salvia* compounds have been demonstrated, including α-glucosidase inhibition (a treatment for diabetes type II) [14], antimicrobial activities [13], anti-inflammatory activities [15], antioxidant quality [13,14], antibacterial activity [16], allelopathic activity [17,18] and anticancer effects [19]. These activities are predominantly correlated with phenolic compounds, such as rosmarinic acid, caffeic acid, salvianolic acid, protocatechuic acid, protocatechualdehyde, luteolin, and rutin [14,15].

Rosmarinic acid is the main phenolic substance in *Salvia* species [20,21]. It has been used as an antiviral (e.g., HIV-1), antibacterial, anti-inflammatory, antithrombotic, and antioxidant compound [22,23]. There are many reports on *Salvia* essential oils, with a high variation in content and composition. According to the review carried out by Hassan-zadeh [24], geraniol, β-caryophyllene, α-pinene, and 1, 8-cineol are the main components of *Salvia* species.

In the 'Flora of Iran', 61 species of *Salvia* have been identified, 17 of which (28%) are endemic [25] in Iran. Based on a literature review, *Salvia* domestication studies are limited to a few aspects. These aspects include autecological properties of *S. hydrangea* L. [26], selection of the best harvesting data in *S. eremophila* Boiss [27], determination of cardinal temperature in *S. hispanica* [28] and genetic diversity of *Salvia miltiorrhiza* [29]. However, limited research has been done on endemic species, and it is very important to pay attention to the morphological aspects and secondary substances of these *Salvia* species in line with the first steps of the domestication of this valuable genus, so that we can be more aware that their selection was used in the production of hybrid cultivars. Among the many reasons for the importance of domestication of medicinal plants is meeting the ever-increasing demand for medicines, particularly for critically endangered species. In addition, it helps conserve the wild genetic diversity of such species. The present study aimed to (a) cultivate 11 *Salvia* genotypes (including eight species) and evaluate adaptability, (b) evaluate some agronomical parameters in association with biomass, oil yield, and species origin, and (c) evaluate and compare the biochemicals of essential oil compounds using multivariate statistical techniques.

## 2. Materials and Methods

### 2.1. Plant Materials

In total, 11 genotypes comprising eight species of *Salvia* were utilized, and seeds were collected from nature (Table S1, Figure 1). The voucher specimens of all species were prepared using two flat presses during the flowering period. All species were authenticated at the Ferdowsi University of Mashhad Herbarium.

### 2.2. Study Site

The study was performed in the experimental fields of the Ferdowsi University of Mashhad in northeastern Iran (36°17′25″ N and 59°35′45″ E; 985 m above sea level). The area is characterized as semi-arid, with hot summers and cold winters. The average climatic data during the growth period (1 May to 31 October) are shown in Table S2. It should be mentioned that the top 0–30 cm of soil was sampled. It was a loamy soil with pH = 7.3 and EC = 2.3 ds/m. The total NPK was 750, 13.5, and 180 ppm, respectively.

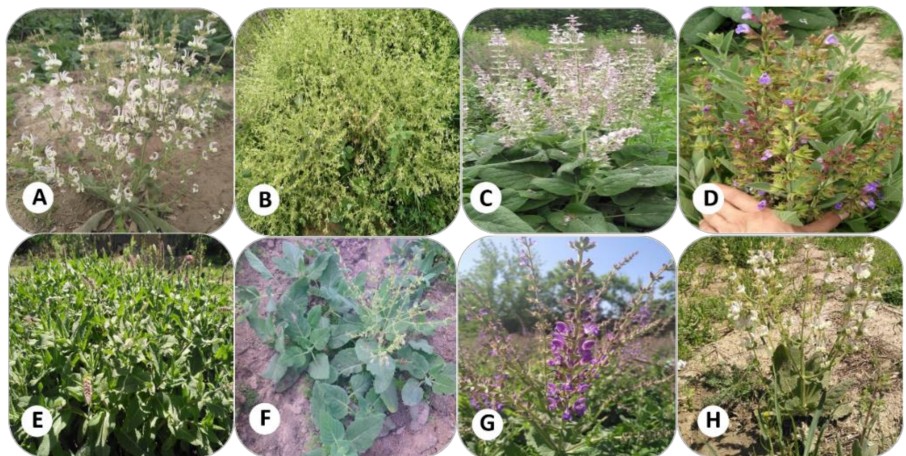

**Figure 1.** *Salvia* species cultivated in farm conditions. (**A**), *S. atropatana*; (**B**), *S. macrosiphon*; (**C**), *S. sclarea*; (**D**), *S. officinalis*; (**E**), *S. nemorosa*; (**F**), *S. syriaca*; (**G**), *S. virgata*; (**H**), *S. frigida*.

### 2.3. Seedling Production

All seeds were washed with running water, primed with 100 ppm $GA_3$ for 24 h, and sown in a greenhouse on 20 January 2016. Primed seeds were grown in 1-L pots containing peat moss and perlite (3:1). The seedlings with 10 leaves were transplanted to the farm on 2 May 2016. In total, 16 seedlings of each genotype were planted in completely randomized block design plots (1.5 m × 1.5 m) with three replications. A furrow system of irrigation was laid out in the studied plots. No herbicides or chemical fertilizers were applied during the experiment, and weeding was performed manually.

### 2.4. Evaluation of Growth Parameters

Plant samples were collected when more than 50% of the plants were blooming. Three plants in each plot for each genotype were harvested during the first hour of the day. The sample fresh weight of the root (RFW), stem (SFW), leaf (LFW), and flower (FFW) were measured separately. Furthermore, the root architecture was evaluated and photographed. The fresh materials were dried at 35 °C, and total dry weight (TDW, g/plant) was recorded. Dry materials were kept in storage at 4 °C for further analysis. The leaf area (LA, $cm^2$/plant) and 1000 seed weight (SW, g) were determined using other plants from each plot.

### 2.5. Evaluation of Biochemical Parameters

The air-dried samples of *Salvia* genotypes (0.5 g) were powdered and used for extraction with 70% methanol (5 mL) for 24 h, kept on a shaker (200 rpm), and subsequently centrifuged at 6000 rpm for 1 min [30]. The supernatants were used for biochemical evaluation and kept in −20 °C.

#### 2.5.1. Total Phenolic Compounds

Total phenolic compound (TP) content was estimated using the Folin–Ciocalteu reagent [30,31]. It should be noted that 100 μL of the methanol extract was diluted using 2 mL deionized water in a 5 mL test tube, and 200 μL of 50% Folin–Ciocalteu reagent was added. After 3 min, 1 mL of 20% (*w/v*) sodium carbonate was added. After 2 h of incubation at 25 °C, the absorption was measured at 765 nm, and compared to a gallic acid calibration curve to estimate the mg of Gallic acid/g extract (Figure S1).

#### 2.5.2. Total Flavonoids

Total flavonoid (TF) contents were determined using a modified colorimetric aluminum chloride method, with quercetin as a standard [30,32]. One mL of extract was mixed with 300 μL of 5% sodium nitrate, 600 μL of 10% aluminum chloride, and 2 mL of 0.5 N sodium hydroxide in a 10 mL test tube, and the volume was increased to 10 mL

with distilled water. The absorption of the reaction mixture was measured at 510 nm and compared to a quercetin standard curve (Figure S2).

### 2.5.3. DPPH Radical Scavenging Activity

The antioxidant activity was measured according to the method described by Brand-Williams [33] with minor modifications [30]. In total, 100μL of the extracts with various concentrations (250–4000 ppm) were added to 5 mL 0.004% 1, 1-diphenyl-2-picrylhydrazyl (DPPH). The samples were incubated in the dark at room temperature for 30 min. The absorption was read spectrophotometrically at 517 nm immediately. The blank sample contained 100 μL methanol plus 5 mL DPPH solution. The antiradical activity was calculated using Equation (1):

$$AA = 1 - (Ab - As)/Ab \times 100 \tag{1}$$

AA: antiradical activity (%); Ab: absorption of the blank; As: absorption of the test sample.

### 2.5.4. Rosmarinic Acid Content

The rosmarinic acid content (RAC) was determined using the method developed [34,35]. 3 mL of 50% methanol were added to 50 mg of fresh weight of each leaf sample. This was incubated at 55 °C for two hours; afterward, 1 mL was diluted with 5 mL of 50% methanol. The diluted extracts were mixed, and absorption was measured at 333 nm. The RA concentration (mg/L) and RAC (mg/g DW) were calculated using Equations (2) and (3), respectively.

$$A = \varepsilon bc \tag{2}$$

A: absorption at 333 nm; $\varepsilon$: extinction coefficient: 19,000 L mol$^{-1}$ cm$^{-1}$; b: width of the disposable cuvettes (1 cm); c: RA concentration;

$$R = \frac{34,105.2 \times A333}{(100 - MC) \times FW} \tag{3}$$

For determining RC on the basis of plant dry weight the following formula was used.
R: RAC; A333: absorption at 333 nm; MC: moisture content; FW: fresh weight (mg), 34,105.2 is a dilution factor.

### 2.5.5. Essential Oil Extraction

Volatile compounds were isolated using the hydro-distillation method [9]. 100 g of dried aerial parts of the plants were extracted using a Clevenger apparatus for four hours. The essential oil content (EOC) was calculated as relative percentage units based on the dry weight (*w/w*). The essential oils were stored in a dark chromatography vial at 4 °C.

### 2.5.6. Gas Chromatography-Flame Ionization Detector and Gas Chromatography-Mass Spectrometry Analysis

Gas chromatography (GC) analysis was performed using Thermo-UFM ultra-fast gas chromatography with an HP-5 fused silica column, 10 m × 0.1 mm i.d., film thickness 0.40 μm, and helium as the carrier gas (32 cm/s). The temperature of the detector and the injector port was 285 °C. Moreover, the oven temperature was initially at 60, which was programmed to increase to 285 °C at the rate of 8 °C/min. Diluted samples (1 μL) were manually injected.

The GC-mass spectrometry [2] analysis was carried out in a Varian 3400 GC-MS system equipped with a DB-5 fused silica column (30 m × 0.25 mm i.d., film thickness: 0.25 μm). It should be noted that helium was used as the carrier gas (30.0 cm/s). The oven temperature was 50–240 °C at a rate of 4 °C/min and the transfer line temperature was 260 °C. The oils were diluted in dichloromethane at various rates (2, 4, and 5 μL of oils in 2 mL solvent), and then 2 μL of each was injected into the GC-MS manually.

A computer library and n-alkanes (C6–C24) were used for compound identification based on GC retention indices. The oil compounds were identified by matching retention indices to Wiley and Adams Mass Spectral libraries, as well as by comparing mass spectra with those published in the literature [36].

*2.6. Statistical Analysis*

The experiment was arranged in a randomized complete block design with three replications (*n* = 3). It is noteworthy that all measurements were made, at least in triplicate. The agronomic and biochemical data analyses were performed using JMP statistical software (version 8.0, SAS Institute, Stockholm, Sweden). The mean values were compared using a one-way ANOVA, followed by Duncan's test ($p \leq 0.05$) [37]. Hierarchical Cluster Analysis (HCA) and Principal Components Analysis (PCA) were used to estimate species similarity according to the essential oil compounds using SAS® (version9.4, SAS Institute, Stockholm, Sweden) and Minitab® (version 18, Penn State University, State College, PA, USA) software. The agro-morphological and biochemical similarity coefficients were determined using Pearson's correlation method at a 95.0% confidence level.

### 3. Results and Discussion

There were significant differences ($p \leq 0.05$) among the species and genotypes regarding all the growth parameters (Tables 1 and S3). For the majority of the studied species, it was noticed that the leaves were produced at the base of the stems. *Salvia sclarea* produced the highest overall biomass (6235 kg/ha), the highest total fresh weight (TFW) (570.2 g/plant) and total dry weight (TDW) (120.5 g/plant), as well as the highest stem, leaf, and flower fresh and dry weights (Table 1). In addition, it had the largest leaf area (9117.3 cm$^2$). The leaf area depends on many factors, such as species, development stage, growth condition, and production management [38]. According to the results of the present study, *S. macrosiphon* had the highest value of the fresh and dry weights of the roots compared to the others.

Significant variations were observed in the root fresh weight (RFW), root dry weight (RDW), and seed weight of *S. virgata* populations (Afoos, Darkesh, and Eghlid). *Salvia virgata* Darkesh population had the lowest leaf area, leaf fresh weight (LFW), and leaf dry weight, compared to the other populations, as it originated from the northwest of Iran with warmer weather.

*S. sclarea* had the lowest RDW:TDW (4.4%) ratio (Figure S3), whereas *S. frigida* (Targavar), *S. frigida* (Jahrom), *S. virgate* (Eghled), and *S. macrosiphon* had the highest RDW:TDW ratio (43.3, 44.6, 46.0, and 44.3%, respectively). *S. nemorosa* and *S. syriaca* showed the highest SDW:TDW (36.2 and 38.3%, respectively), and the lowest values were recorded in *S. sclarea* (17.2%). The highest LDW:TDW values were recorded in *S. sclarea* and *S. officinalis*, and *S. virgata* (Darkesh) (65.6 and 55.9%) was lowest (18%). *S. nemorosa* had the highest FDW/TDW and *S. atropatana* and *S. frigida* (Jahrom) were lowest (5.9 and 4.9%).

*Salvia frigida* Targavar population had a lower RDW:TDW ratio (43.3%) compared to the Jahrom population (44.6%). Jahrom population originates from the near south of Iran and is exposed to warm conditions; accordingly, it showed a lower leaf area (1025 cm$^2$/plant). Targavar population genetically is a small compact plant consistent with its origin from the northeast of Iran with a high altitude and cold climate.

Root architecture, such as the increase in the number of small roots of *S. nemorosa* (Figure 2b) to provide more absorption surface, and the presence of specialized tissues, such as rhizome in *S. syriaca* and semi-tuberous roots in *S. macrosiphon* (Figure 2g), also affect the ability of plants to absorb water. Root morphology depends on many factors, such as species, whereas the root system is highly variable and controlled by plant genetics and environmental conditions [39].

**Table 1.** Agronomical parameters of various *Salvia* species after two years of cultivation [1].

| Species | LA [1] | TFW | TDW | RFW | RDW | SFW | SDW | LFW | LDW | FFW | FDW | SW | TDB |
|---|---|---|---|---|---|---|---|---|---|---|---|---|---|
| *S. atropatana* | 547.7 cd | 91.7 d | 22.4 ef | 25.0 efg | 5.3 ef | 33.3 d | 6.9 d | 51.7 bcd | 8.9 cd | 6.2 d | 1.3 f | 4.20 b | 1182 cd |
| *S. macrosiphon* | 604.0 cd | 144.2 c | 52.7 b | 107.7 a | 23.2 a | 60.7 bc | 13.0 bc | 54.0 bcd | 10.7 bc | 29.0 b | 5.8 c | 2.65 d | 1764 c |
| *S. sclarea* | 9117.3 a | 570.2 a | 120.5 a | 25.7 efg | 5.3 f | 96.0 a | 20.7 a | 409.7 a | 79.1 a | 64.0 a | 15.4 a | 2.24 f | 6235 a |
| *S. officinalis* | 1128.3 c | 116.8 cd | 25.8 ef | 13.3 g | 3.2 f | 31.3 d | 6.3 d | 73.0 b | 14.4 b | 12.0 cd | 1.9 ef | 5.51 a | 1452 cd |
| *S. nemorosa* | 1936.3 b | 186.4 b | 43.6 bc | 32.5 def | 6.7 def | 74.9 b | 15.8 b | 55.5 bc | 10.5 bc | 55.5 a | 10.6 b | 0.77 j | 3288 b |
| *S. syriaca* | 348.0 d | 88.5 d | 19.3 f | 21.fg | 3.4 f | 42.7 cd | 7.4 d | 28.3 cd | 5.5 cd | 18.9 bcd | 3.0 def | 3.65 c | 1014 d |
| *S. virgata (Afoos)* | 839.7 cd | 101.7 d | 30.6 de | 46.8 de | 10.1 cd | 41.0 d | 9.1 cd | 36.5 cd | 8.0 cd | 23.8 bc | 3.4 c.f | 1.15 h | 1344 cd |
| *S. virgata (Darkesh)* | 760.0 cd | 89.5 d | 27.4 ef | 42.8 cde | 10.0 cde | 37.3 d | 7.9 d | 25.7 d | 4.9 d | 26.0 b | 4.5 cd | 0.95 i | 1110 cd |
| *S. virgata (Eghled)* | 926.8 cd | 119.4 cd | 38.9 cd | 78.2 b | 17.9 b | 45.0 cd | 8.6 cd | 50.5 bcd | 8.6 cd | 23.5 bc | 3.8 cde | 1.59 g | 1368 cd |
| *S. frigida (Jahrom)* | 1025.0 cd | 182.2 b | 28.7 ef | 58.4 c | 11.7 c | 27.3 d | 6.0 d | 56.3 bcd | 9.5 bcd | 9.0 d | 1.5 ef | 2.42 e | 924 d |
| *S. frigida (Targavar)* | 1073.3 c | 81.7 d | 22.8 ef | 40.8 cde | 7.0 def | 26.7 d | 5.4 d | 43.2 cd | 8.2 cd | 11.3 cd | 2.2 def | 2.48 e | 1230 cd |

Values were expressed as means ± SD (n = 3). LA, Leaf area (cm$^2$/plant); TFW, Total fresh weight (g/plant); TDW, Total dry weight (g/plant); RFW, Root fresh weight (g/plant); RDW, Root dry weight (g/plant); SFW, Stem fresh weight (g/plant); SDW, Stem dry weight (g/plant); LFW, Leaf fresh weight (g/plant); LDW, Leaf dry weight (g/plant); FFW, Flower fresh weight (g/plant); FDW, Flower dry weight (g/plant); SW, 1000 Seeds weight (g); TDB, Total dry biomass (Kg/ha). Values followed by the same letter (a, b, c, d, e, . . . ) are not significantly different ($p < 0.05$) by Duncan's multiple range test.

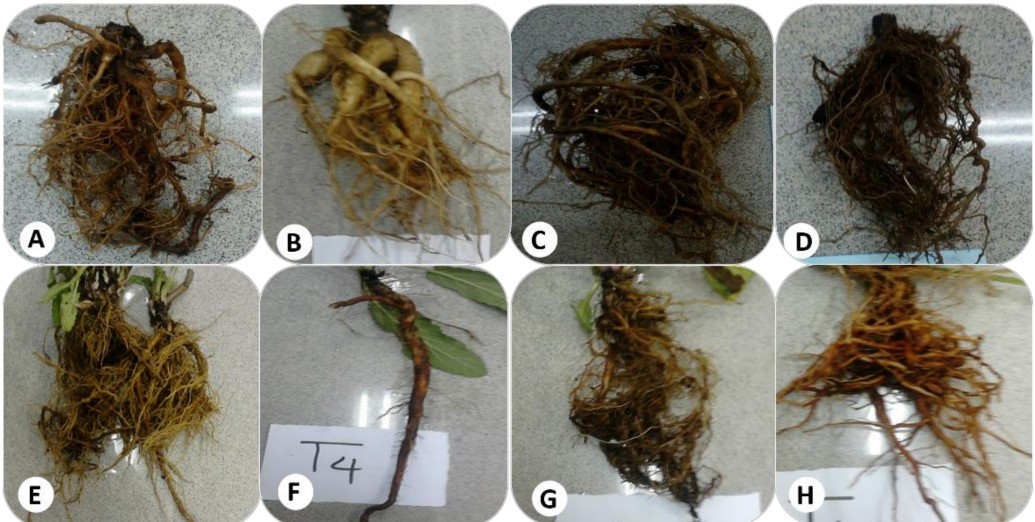

**Figure 2.** Root system of five-month-old *Salvia* species. (**A**), *S. atropatana*; (**B**), *S. macrosiphon*; (**C**), *S. sclarea*; (**D**), *S. officinalis*; (**E**), *S. nemorosa*; (**F**), *S. syriaca*; (**G**), *S. virgata*; (**H**), *S. frigida*.

### 3.1. Phytochemical Evaluation

The analysis of the biochemical compounds in various plant organs showed that leaf and flower parts had the highest bioactive compounds (Table 2). The contents of total phenolic compounds, total flavonoids, and antioxidant activity were determined for leaf, root, and some stem and flower samples of the *Salvia* species (Table 2). There was a significant variation in the biochemical compounds among the genotypes ($p \leq 0.05$).

The biochemical compounds usually varied between plant organs, within harvesting stages [40], and even during the harvesting time during the day as well as plant nutrition [9,41], post-harvest operations [13,20,42]. In medicinal plants, genetics and plant breeding aim to improve the proportion of the utilized plant parts (e.g., leaves and flowers) in the total plant mass [42,43]. Phenolic and flavonoid compounds have important pharmacological activities, including antioxidant effects (in some cases, more than vitamins C and D), anti-tumor, and antibacterial effects [22,44]. For example, strong antibacterial activity in *S. brachyantha*, *S. microstegia*, and *S. aethiopis* against three gram-positive bacteria was reported by Tohma [13]. Shaerzadeh [45] reported the protective effect of *S. sahandica* extract on neurons through antioxidant and anti-glycogenic activity.

#### 3.1.1. Total Phenolic Compounds (TP)

For most genotypes, the highest concentration of TP was found in the leaves (10.4–40.6 mg GAE/g extract) compared to roots (1.5–18.5 mg GAE/g extract); however, the flowers of several species also contained high levels of TP, but the TP concentration in the stem was low (4.2–18.7 mg GAE/g extract). The highest concentrations of TP were observed in *S. officinalis* leaf (40.6 mg GAE/g extract), *S. nemorosa* flower (39.8 mg GAE/g extract), *S. virgata* (Afoos) leaf and flower (35.6 and 36.7 mg GAE/g extract, respectively), and *S. virgata* (Darkesh) flower (32.5 mg GAE/g extract).

Given the higher proportion of DW in the leaves, *S. officinalis* and *S. virgata* (Afoos) had the highest potential yield of TP. The aerial parts of *Salvia* species (especially their leaves and flowers) are densely covered with trichomes, which play a key role in the synthesis and storage of biochemical compounds, such as phenolic compounds [46]. At the plant cell level, phenolic compounds are usually present in the vacuoles of colored tissue, such as leaves and flower petals [42,47]. However, in some plants, such as *Echinacea purpurea*, phenolic compounds are mainly distributed in the root cortex, phloem parenchyma cells, vascular rays, and pits. Phenylalanine ammonia-lyase (PAL) is a key enzyme in the shikimate pathway, which is mainly located in subepidermal cells and vascular parenchyma cells in plants [47]. It eliminates ammonia from phenylalanine to form *trans*-cinnamic acid, a

precursor of lignins, flavanoids, and coumarins [48]. PAL plays an important role in plant resistance to various stressors (biotic and abiotic), and by increasing Pal gene expression, it will be possible to improve stress resistance in plants [49].

**Table 2.** DPPH assay and total phenolic compound and flavonoid content of various *Salvia* genotypes organs.

| Species | Plant Parts | Total Phenolic Compounds (mg GAE/g Extract) | Flavonoids Content (mg QE/g Extract) | IC50 (1) |
|---|---|---|---|---|
| *S. atropatana* | Leaf | 22.7 ± 3.9 d–g | 23.9 ± 2.9 lm | 14.0 ± 1.1 ij |
| | Root | - | - | - |
| | Stem | - | - | - |
| | Flower | - | - | - |
| *S. macrosiphon* | Leaf | 14.7 ± 2.0 g–k | 51.9 ± 5.5 hij | 36.7 ± 1.9 e |
| | Root | 5.8 ± 0.8 j–m | 15.9 ± 1.9 ms | 10.9 ± 1.7 j |
| | Stem | - | - | - |
| | Flower | - | - | - |
| *S. sclarea* | Leaf | 19.3 ± 2.3 fi | 69.9 ± 3.2 fgh | 24.2 ± 1.8 fg |
| | Root | 1.5 ± 0.2 m | 14.9 ± 2.1 m | 14.9 ± 2.0 ij |
| | Stem | - | - | - |
| | Flower | - | - | - |
| *S. officinalis* | Leaf | 40.6 ± 2.9 a | 129.1 ± 7.9 b | 52.3 ± 3.1 c |
| | Root | 4.9 ± 0.3 klm | 96.3 ± 8.4 cde | 42.9 ± 7.0 d |
| | Stem | - | - | - |
| | Flower | - | - | - |
| *S. nemorosa* | Leaf | 29.0 ± 2.3 c–f | 141.9 ± 10.0 ab | 90.6 ± 3.3 a |
| | Root | 7.3 ± 0.9 j–m | 71.9 ± 8.1 fh | 24.3 ± 0.8 fg |
| | Stem | 9.7 ± 2.1 i–m | 68.8 ± 3.0 ghi | 14.3 ± 1.2 ij |
| | Flower | 39.8 ± 6.9 ab | 121.7 ± 4.8 bc | 90.7 ± 7.2 a |
| *S. syriaca* | Leaf | 13.7 ± 1.4 g–l | 43.6 ± 6.8 i–l | 52.0 ± 2.5 c |
| | Root | - | - | - |
| | Stem | - | - | - |
| | Flower | - | - | - |
| *S. virgata* (Afoos) | Leaf | 35.6 ± 7.6 abc | 83.9 ± 9.4 d–g | 89.1 ± 1.4 a |
| | Root | 15.8 ± 3.3 g–j | 94.9 ± 4.7 def | 38.2 ± 5.3 de |
| | Stem | 18.7 ± 2.6 fi | 82.1 ± 7.3 d–g | 42.2 ± 3.6 de |
| | Flower | 36.7 ± 5.7 abc | 100.4 ± 8.7 cd | 89.6 ± 3.0 a |
| *S. virgata* (Darkesh) | Leaf | 14.2 ± 2.3 g–l | 54.3 ± 7.7 hij | 90.3 ± 2.2 a |
| | Root | 18.5 ± 0.3 ghi | 63.5 ± 4.3 ghi | 25.3 ± 1.7 fg |
| | Stem | - | - | - |
| | Flower | 32.5 ± 2.7 ad | 156.7 ± 6–5 a | 53.8 ± 1.6 c |
| *S. virgata* (Eghled) | Leaf | 18.3 ± 3.4 ghi | 61.5 ± 8.8 ghi | 22.4 ± 4.1 gh |
| | Root | 10.2 ± 1.3 i–m | 71.7 ± 2.7 e–h | 28.3 ± 1.4 f |
| | - | - | - | - |
| | - | - | - | - |
| *S. frigida* (Jahrom) | Leaf | 10.4 ± 0.5 i–m | 50.9 ± 7.0 hk | 29.1 ± 2.3 f |
| | Root | 11.1 ± 0.7 hm | 34.0 ± 4.7 j–m | 18.0 ± 2.6 hi |
| | Stem | - | - | - |
| | Flower | - | - | - |
| *S. frigida* (Targavar) | Leaf | 30.1 ± 1.5 b–e | 95.1 ± 5.3 def | 88.5 ± 2.3 a |
| | Root | 10.1 ± 24 i–m | 62.5 ± 4.7 ghi | 26.1 ± 2.6 fg |
| | Stem | 4.2 ± 0.8 lm | 26.3 ± 3.0 klm | 40.2 ± 3.6 de |
| | Flower | 21.1 ± 2.6 e–h | 58.8 ± 8.7 g–j | 81.0 ± 4.3 b |

Values followed by the same letter (a, b, c, d, e, … ) are not significantly different ($p < 0.05$) by Duncan's multiple range test.

Loizzo et al. reported a similar range of TP and flavonoid concentrations among nine *Salvia* species: 5.1–42.5 mg GAE/g and 2.2–36.2 mg (+)-catechin equivalents per g of dry extract for *S. xanthocheila* and *S. glutinosa*, respectively [50]. Bahadori [22] stated much higher TF content for dichloromethane and methanol extracts of *S. syriaca* (67–255 mg GAE/g); however, they collected their samples from natural pastures, where the plant was exposed to various sources of stress. These differences are probably the result of variations in locality, climatic conditions, environmental stress, seasonal factors, and solvent types [51].

Hamrouni-Sellami [52] reported a total phenolic compound range of 0.399–2.337 mg GAE/g DM in *S. officinalis* dried using different methods. Moreover, there is no consistent amount of total phenolic compounds in plants; it varies between species, even within the same genre. Evidence from several studies suggests that *S. officinalis* has potent antioxidant activity. The phenolic compounds are isolated from the extract of *Salvia officinalis* L., such as carnosol, rosmarinic acid, and carnosic acid, followed by caffeic acid, rosmanol, rosmadial, genkwanin, and cirsimaritin as flavonoid compounds, which have the most effective antioxidant activity [53,54]. Koşar et al. [31] reported TP (28.3–212.3 GAE/g extract) in various extracts of *S. virgata* aerial parts.

### 3.1.2. Total Flavonoids

As shown in Table 2, *S. officinalis* and *S. nemorosa* had the highest concentrations of TF in their leaves (129.1 and 141.9 mg QE/g extract, respectively), while *S. nemorosa* and *S. virgata* (Darkesh) had similar high concentrations in their flowers (121.7 mg QE/g extract and 156.7 mg QE/g extract, respectively). The minimum amount of flavonoids was obtained from the extract of *S. sclarea* (14.9 mg QE/g extract) and *S. macrosiphon* (15.9 mg QE/g extract). There were usually significantly higher concentrations of TF in the leaves compared to the roots except for *S. virgata* genotypes, where the concentrations were similar in both tissues (root: 71.7 mg QE/g extract, leaf: 61.5 mg QE/g extract). The *Salvia* sclarea had lower concentrations of TF (less than half the above), but several-fold biomasses (6235 kg dry biomass per ha) (Table 2) and therefore greater potential total yield.

According to previous studies, flavonoids are stored in the epidermis and cuticle cells of plant leaves [42]. At the plant subcellular level, most flavonoid-synthesizing enzymes are located in the endoplasmic reticulum, where the pigments themselves accumulate in the vacuole [55].

Bahadori [22] reported that the total flavonoids of different extracts of *S. syriaca* ranged from 83 to 127 mg QE/g extract, which is comparable to the results of the present study. However, Loizzo [50] reported much lower concentrations in *S. sclarea* (12.2 mg (+)-catechin equivalents/g extract). Nevertheless, according to the research performed by Lee [56], the difference in the amount of flavonoids depends on many factors, such as harvest time or extraction methods.

### 3.1.3. DPPH Radical Scavenging Activity

The highest radical scavenging activity was in roots of *S. macrosiphon* and *S. sclarea* (IC50 = 10.9 and 14.9 µg/mL, respectively), leaves of *S. atropatana* (IC50 = 14.0 µg/mL), and stem of *S. nemorosa* (IC50 = 14.3 µg/mL) (Table 2). In contrast, the leaf extract of *S. nemorosa* (IC50 = 90.6 µg/mL), *S. virgata* (Afoos) (IC50 = 89.1 µg/mL) and (Darkesh) (IC50 = 90.3 µg/mL) and flower extract of *S. nemorosa* (IC50 = 90.7 µg/mL) and *S. virgata* (Afoos) (IC50 = 89.6 µg/mL) had substantially lower antiradical activities which can be linked to the TP contents.

These findings are consistent with those of previous studies. For example, Bahadori [22] reported various levels of antioxidant activity in different extracts (IC50 = 70–245 µg/mL in methanol and dichloromethane extracts) of *S. syriaca* collected from the northwest of Iran. The antioxidant activity of nine *Salvia* species was much higher, with an IC50 of 3.2–17.5 µg/mL in *S. atropatana* and *S. glutinosa*, respectively [50]. In other antioxidant studies of *Salvia* species, the IC50 in the root samples of *S. miltiorrhiza* and *S. verticillata* was less than that of the leaf [11]. Phenolic compounds and diterpenoids are two major

groups of compounds that are postulated in *Salvia* antioxidant activity [11]. For example, spathulenol has a high antioxidant activity reported in various *Salvia* species [22]. Therefore, antioxidant activity in *Salvia* plant parts varies widely and depends on the biochemical content of the organs.

The variation in TP, TF, and IC50 was significant among the species and the organs (root, leaf, stem, and flower). It has been established that different factors, such as plant genetics, agronomical practices [9], harvesting time and drying, extraction procedure, and solvent, affect the yield of medicinal compounds of cultivated herbs [6,13,57,58].

### 3.1.4. Rosmarinic Acid Content

There were significant differences in rosmarinic acid content (RAC) among species, ranging from 0.28 to 0.47 mg/g DW (Figure 3). Four species had more than 0.40 mg/g DW. The RAC in *Salvia* species was reported with a high variation in previous research [9,20,22]. Rosmarinic acid is a simple depside formed by the condensation of two single phenolic acids, one derived from the phenylpropanoid pathway and the other from the tyrosine-derived pathway [19].

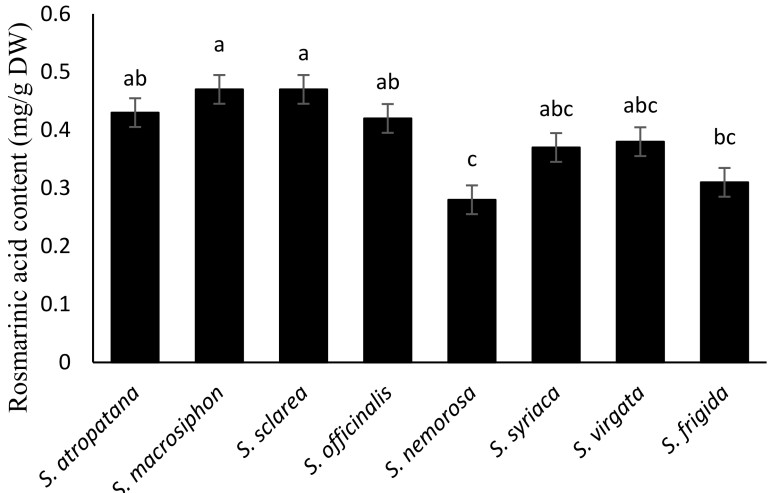

**Figure 3.** Comparison of rosmarinic acid content in leaves of *Salvia* species. Different letters over bar indicates significant difference in treatment according to Duncan's multiple range test ($p < 0.05$).

According to Zhang [58], rosmarinic acid and salvianolic acid B were the main phenolic acids in the leaf extract of *S. miltiorrhiza*. Quantitative analysis of phenolic compounds in *S. sahandica* using the high-pressure liquid chromatography method showed that rosmarinic acid (67.12 mg/g extract) was the most important phenolic compound in this plant [45]. In other species, such as *S. brachyantha*, rosmarinic acid has been introduced as the main phenolic compound [7,13]. Analysis of phenolic compounds by HPLC in *S. officinalis* flower extract showed that rosmarinic acid is the most abundant phenolic compound in this plant [56]. A similar study on other species of this genus indicated its importance and quantity. Therefore, it seems that *Salvia* species are a valuable source of rosmarinic acid and could be considered in the pharmaceutical and food industry.

### 3.1.5. Essential Oil Content and Composition

The essential oil yield varied from 0.35% in *S. atropatana* to 1.45% (*v/w*) in *S. officinalis* (Table 3). The yield of medicinal plants' essential oil depends on genetic background, environmental conditions, edaphic factors, harvesting time and phenological stages of plants, and drying process [20,41,59]. Predominantly, the essential oils of most plants consist of two or three main compounds. In total, 60 compounds were identified (Table 3).

**Table 3.** Essential oil composition (% of principal components) of some *Salvia* species cultivated in Khorasan Razavi province, Mashhad—Iran.

| No. | Formula | Compounds | RI [1] | SO [2] | SV [3] | SN [4] | SS [5] | SC [6] | SM [7] | SA [36] | SF [9] |
|---|---|---|---|---|---|---|---|---|---|---|---|
| 1 | C10H16O | α-Thujone | 1105 | 39.34 | - | - | - | - | - | - | - |
| 2 | C10H16O | β-Thujone | 1112 | 15.51 | - | - | - | - | - | - | - |
| 3 | C10H16O | (-)-Camphor | 1139 | 17.18 | - | - | 15.12 | - | 3.10 | - | - |
| 4 | C11H18O2 | Isobornyl formate | 1217 | 4.60 | - | - | 18.40 | - | - | - | - |
| 5 | C10H14 | 1,3,8-p-menthatriene | 1100 | 2.98 | - | - | 5.07 | - | - | - | - |
| 6 | C10H16O | trans-3-Caren-2-ol | 1655 | 0.67 | - | - | 4.17 | - | - | - | - |
| 7 | C10H14 | 2,8-Decadiyne | 1102 | 0.47 | - | - | - | - | - | - | - |
| 8 | C10H16O | Pulegone | 1234 | 1.34 | - | - | - | - | - | - | - |
| 9 | C10H16 | Camphene | 946 | 0.31 | - | - | 5.39 | - | - | - | - |
| 10 | C15H24 | Aromadendrene | 1442 | 1.33 | 0.48 | 47.50 | - | - | 2.88 | - | - |
| 11 | C15H24 | α-Caryophyllene | 1438 | 13.03 | 0.50 | - | - | - | - | 0.50 | - |
| 12 | C10H14O | Thymol | 1291 | - | 0.34 | - | - | 0.74 | - | - | 3.84 |
| 13 | C15H24 | β-Caryophyllene | 1441 | - | 7.08 | - | - | 6.18 | 13.50 | - | 31.05 |
| 14 | C15H24 | cis-β-Famesene | 1458 | - | 1.46 | - | - | - | - | - | - |
| 15 | C15H24 | δ-Cadinene | 1525 | - | 23.32 | - | - | - | - | 1.55 | - |
| 16 | C15H24 | Seychellene | 1449 | - | 1.20 | - | - | - | - | - | - |
| 17 | C15H24 | β-Elemen | 1391 | - | 0.96 | - | - | - | - | - | - |
| 18 | C15H24 | cis-Z-α-Bisabolene epoxide | 1816 | - | 2.73 | - | - | - | - | - | - |
| 19 | C22H32O2 | Doconexent | 2522 | - | 3.97 | - | - | - | - | - | - |
| 20 | C17H24O | Falcarinol | 2040 | - | 2.07 | - | - | - | - | - | - |
| 21 | C15H24 | Υ-Gurjunene | 1470 | - | 5.62 | - | - | - | - | 0.41 | - |
| 22 | C15H26O | Valeranone | 1672 | - | 26.09 | - | - | - | - | - | 0.60 |
| 23 | C21H34O2 | 3-Ethyl-3-hydroxyandrostan-17-one | 2651 | - | 10.22 | - | - | - | - | - | - |
| 24 | C7H16 | 2,3-Dimethylpentane | 675 | - | - | 4.76 | - | - | - | - | - |
| 25 | C7H16 | 3-Methylhexane | 671 | - | - | 0.34 | - | - | - | - | - |
| 26 | C8H18 | Tetramethylbutane | 716 | - | - | 18.12 | - | - | - | - | - |
| 27 | C18H31N | 2-Tridecylpyridine | 1920 | - | - | 4.21 | - | - | - | - | - |
| 28 | C10H17Br | Geranyl bromide | 1284 | - | - | 1.28 | - | - | - | - | - |
| 29 | C10H16 | Santolina triene | 911 | - | - | 3.72 | - | - | - | - | - |
| 30 | C15H24O | Caryophyllene oxide | 1581 | - | - | 13.48 | - | 0.55 | 14.63 | 24.30 | 5.5 |
| 31 | C20H24N2O | Eseroline, benzyl ether | 2635 | - | - | 2.08 | - | - | - | - | - |
| 32 | C10H16 | α-Pinene | 937 | - | - | - | - | - | - | - | 6.15 |
| 33 | C10H18O | α-Terpineol | 1192 | - | - | - | 5.99 | 3.21 | 0.21 | - | 3.16 |
| 34 | C10H16O | cis-Verbenol | 1115 | - | - | - | 13.55 | - | - | - | - |
| 35 | C12H20O2 | Bornyl acetate | 1288 | - | - | - | 30.83 | - | - | 5.55 | 5.12 |
| 36 | C10H16 | Myrcene | 991 | - | - | - | - | 1.85 | - | - | - |
| 37 | C10H16 | transe-β-Ocimene | 1039 | - | - | - | - | 1.82 | - | - | - |
| 38 | C10H16 | Terpinolene | 1089 | - | - | - | - | 0.68 | 0.85 | - | 0.55 |
| 39 | C10H18O | Linalool | 1099 | - | - | - | - | 26.20 | 27.20 | 7.60 | - |
| 40 | C10H18O | Nerol | 1229 | - | - | - | - | 2 | - | - | - |
| 41 | C10H20O2 | Linalyl acetate | 1248 | - | - | - | - | 20.50 | 1.55 | - | 1.05 |
| 42 | C15H24 | β-Cubebene | 1387 | - | - | - | - | 0.65 | - | - | - |
| 43 | C15H24 | Germacrene D | 1583 | - | - | - | - | 16.40 | 7.59 | 10.50 | 19.5 |
| 44 | C15H26O | β-Eudesmol | 1651 | - | - | - | - | 0.71 | - | - | - |
| 45 | C10H18O | 1,8-cineole | 1035 | - | - | - | - | - | - | - | 2.70 |
| 46 | C10H16 | Limonene | 1029 | - | - | - | - | - | 0.22 | - | 1.01 |
| 47 | C10H18O | Borneol | 1169 | - | - | - | - | - | 0.77 | - | - |
| 48 | C15H24O | Spathulenol | 1578 | - | - | - | - | - | 5.80 | - | 0.87 |
| 49 | C15H24 | Germacrene B | 1552 | - | - | - | - | - | 1.32 | - | - |
| 50 | C15H26O | α-Eudesmol | 1653 | - | - | - | - | - | 1.25 | - | - |
| 51 | C27H56 | Heptacosane | 525 | - | - | - | - | - | 0.30 | - | - |
| 52 | C15H24 | α-Cubebene | 1353 | - | - | - | - | - | - | 16.4 | - |
| 53 | C20H36O2 | Sclareol | 2223 | - | - | - | - | - | - | 0.50 | - |
| 54 | C17H28O2 | Caryophyllene acetate | 1692 | - | - | - | - | - | - | 3.21 | - |
| 55 | C15H24O | Alloaromadendrene oxide | 1641 | - | - | - | - | - | - | 0.74 | - |
| 56 | C12H20O7 | Triethyl citrate | 1657 | - | - | - | - | - | - | 0.60 | - |
| 57 | C15H24 | Diepicedrene | 1402 | - | - | - | - | - | - | 0.81 | - |
| 58 | C15H24O | 14-Hydroxy-α-humulene | 1715 | - | - | - | - | - | - | 2.61 | - |
| 59 | C14H12O2 | Benzyl benzoate | 1763 | - | - | - | - | - | - | 0.45 | - |
| 60 | C10H8 | Naphthalene | 1185 | - | - | - | - | - | - | - | 1.25 |
| | | Oxygenated monoterpenes | | 78.64 | 0.34 | 0.0 | 88.06 | 52.65 | 32.83 | 12.34 | 15.87 |
| | | Monoterpene hydrocarbons | | 3.76 | 0.0 | 5.0 | 10.46 | 4.35 | 1.07 | 0.81 | 8.96 |
| | | Sesquiterpene hydrocarbons | | 14.36 | 40.62 | 47.5 | 0.0 | 23.23 | 25.29 | 29.36 | 50.55 |
| | | Oxygenated sesquiterpenes | | 0.0 | 28.82 | 13.48 | 0.0 | 1.26 | 21.68 | 31.67 | 6.97 |
| | | Others | | 0.0 | 16.26 | 29.51 | 0.0 | 0.0 | 0.3 | 1.55 | 0.0 |
| | | Detected compound | | 96.76 | 86.04 | 95.49 | 98.52 | 81.49 | 81.17 | 75.73 | 82.35 |
| | | Essential oil content | | 1.45 | 0.70 | 0.85 | 0.45 | 0.85 | 0.55 | 0.35 | 0.40 |

[1] Retention indices; [2] *S. officinalis*; [3] *S. virgata*; [4] *S. nemorosa*; [5] *S. syriaca*; [6] *S. sclarea*; [7] *S. macrosiphon*; [36] *S. atropatana*; [9] *S. frigida*.

In different *Salvia* species, the main components of essential oil were completely different and as follows: α-thujone (39.34%), camphor (17.18%), and β-thujone (15.51%) in *S. officinalis*; aromadendrene (47.5%) and tetramethylbutane (18.12%) in *S. nemorosa*; linalool (26.20%) and linalyl acetate (20.50%) in *S. sclarea*; linalool (27.20%) and caryophyllene oxide (14.63%) in *S. macrosiphon*; caryophyllene oxide (24.30%) and α-cubebene (11.40%) in *S. atropatana*; β-caryophyllene (28.75%) in *S. frigida*, valeranone (26.09%) and δ-cadinene (23.32%) in *S. virgate*, and bornyl acetate (30.83%), isobornylformate (18.4%), camphor (15.12%), cis-verbenol (13.55) in *S. syriaca*.

Oxygenated monoterpenes were the dominant compounds in *S. officinalis* (78.64%), *S. syriaca* (88.06%), *S. sclarea* (52.65%), and *S. macrosiphon* (32.83%). The highest level of oxygenated sesquiterpenes was detected in *S. atropatana* (31.67%). However, due to the low essential oil content in the majority of *Salvia* species, the extraction of the essential oil was performed for more than 3 h, and some other heavy compounds, such as high hydrocarbons and fatty acids, were observed in the essential oil samples.

These results are in line with those of the study performed by Said-Al Ahl [60] who reported that camphor, α-thujone, and sclareol were the main compounds of *S. officinalis* grown in Egypt. A comparison of six *S. nemorosa* populations by Mahdieh et al. showed a wide variation in essential oil compounds [6]. Sefidkon and Mirza [61] found that non-cultivated *S. syriaca* had germacrene B and germacrene D as the main compounds and β-caryophyllene and germacrene D were the major constituents of *S. virgata*.

A comparison of oil samples in the current study with previous publications indicated that the main compounds were the same, but the overall composition varied within and between *Salvia* species. The current study revealed that β-caryophyllene, caryophyllene oxide, and germacrene D are most frequent in the oil of *Salvia* species (Table 3). However, Hassanzadeh [24] reported that geraniol, β-caryophyllene, α-pinene, and 1,8-cineol are the most plentiful across the genus. Numerous factors may contribute to this variation, including genotype, geographic conditions, plant growth stage, and extraction procedure [6,20]. For example, in investigating the methods of extracting essential oil from 10 *Salvia* species, significant variation was detected in their content and composition [62].

The genetic component of the essential oil composition among *Salvia* species is illustrated by HCA (Figure 4). The species were divided into five clusters at a minimum 42.71 similarity levels. The highest similarity was observed in *S. sclarea* and *S. macrosiphon*, and these two species had high amounts of linalool, germacrene D, and β-caryophyllene. *S. nemorosa* was separated from other species; the main constituents, aromadendrene and tetramethylbutane (65.7% of the total), were not observed in the other species.

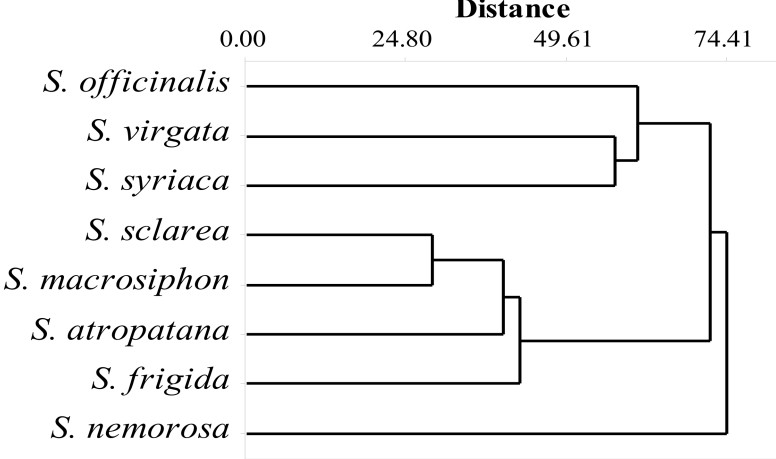

**Figure 4.** Dendrogram obtained by cluster analysis of the essential oil compounds for eight *Salvia* species (Ward method).

According to pairwise Euclidean distance, *S. sclarea* and *S. macrosiphon* were the closest species (with a 28.8 dissimilarity coefficient), while *S. officinalis* and *S. nemorosa* had the least similarity (70.6). Table 4 summarizes the ratio of similarities of all species to each other.

**Table 4.** Pairwise Euclidean distance based on chemical composition data.

| *Salvia* **Species** | *S. atropatana* | *S. macrosiphon* | *S. sclarea* | *S. officinalis* | *S. nemorosa* | *S. syriaca* | *S. virgata* | *S. frigida* |
|---|---|---|---|---|---|---|---|---|
| *S. atropatana* | 0.0 | | | | | | | |
| *S. macrosiphon* | 32.4 | 0.0 | | | | | | |
| *S. sclarea* | 41.8 | 28.8 | 0.0 | | | | | |
| *S. officinalis* | 57.9 | 58.5 | 61.0 | 0.0 | | | | |
| *S. nemorosa* | 57.0 | 58.4 | 65.2 | 70.6 | 0.0 | | | |
| *S. syriaca* | 50.5 | 54.5 | 56.6 | 58.3 | 68.1 | 0.0 | | |
| *S. virgata* | 49.5 | 50.0 | 52.8 | 61.0 | 65.0 | 57.0 | 0.0 | |
| *S. frigida* | 41.1 | 36.7 | 41.4 | 60.3 | 52.9 | 41.4 | 36.7 | 0.0 |

Principle component analysis (PCA) provided more detail about the relationships between *Salvia* species based on essential oil data (Figure 5). The vectors show various components that are present at more than 5.0% in at least one species. As shown in Figure 4, four widely separated groups of species were categorized: *S. virgata* was separated from other species by principal component 1 (PC1), whereas *S. syriaca* was separated by principle component 2 (PC2). The high overlap of *S. atropatana*, *S. macrosiphon*, *S. nemorosa*, *S. frigida*, and *S. sclarea* indicated that these five species had similar volatile profiles. In addition, the essential oil of *S. officinalis* was clustered with a negative value of PC1 and PC2.

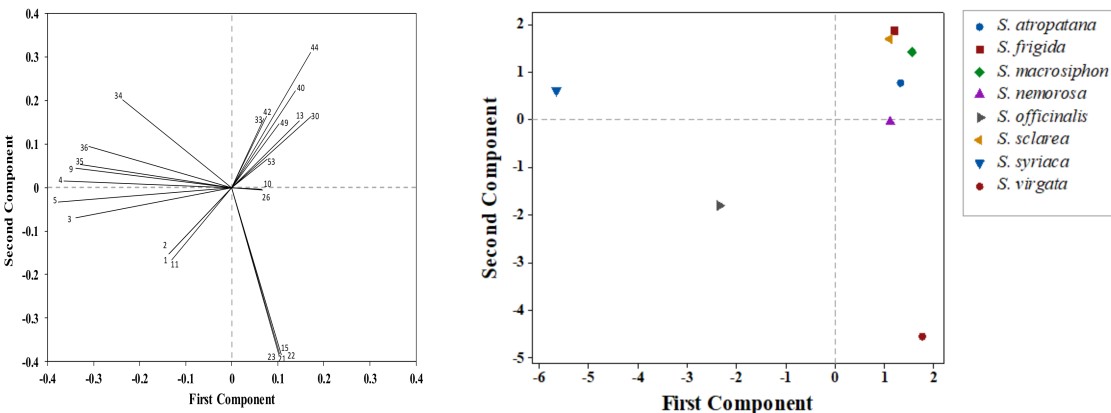

**Figure 5.** Biplot based on the first two principal components of chemical composition data, demonstrating the relationship among compounds identified in the essential oils (**Left**) and its ratios in studied species (**Right**).

### 3.2. Correlations among Traits

There were some positive and negative relationships between the morphological and biochemical parameters (Table 5). The DPPH scavenging assay correlated with TP and TF (r = 0.42 and 0.56, respectively). This suggests that phenolic compounds play an important role in antioxidant activity. Similar correlations were reported by Matkowski [11] in S. *przewalskii*, *S. miltiorrhiza*, and *S. verticillate* [9], by Zhang [58] in *S. miltiorriza*, and by Bahadori [22] in *S. syriaca*. Phenolic compounds substantially scavenge free radicals through chelating (bonding with metal ions) and the donation of electrons [50,57]. The essential oil content had a positive correlation with TF (r = 0.49). On the other hand, agronomical characteristics did not correlate with EOC. There was a positive correlation between agronomical traits ranging from r = 0.58 to r = 0.98 except with RFW and RDW.

**Table 5.** Pearson's correlation coefficient of various agronomical and biochemical traits studied among *Salvia* species [1] (Pl).

| | LA | TFW | TDW | RFW | RDW | SFW | SDW | LFW | LDW | FFW | FDW | SW | IC50 | TP | TF | RAC | EOC |
|---|---|---|---|---|---|---|---|---|---|---|---|---|---|---|---|---|---|
| LA | 1 | | | | | | | | | | | | | | | | |
| TFW | 0.98 ** | 1 | | | | | | | | | | | | | | | |
| TDW | 0.97 ** | 0.98 ** | 1 | | | | | | | | | | | | | | |
| RFW | −0.2 | −0.12 | −0.14 | 1 | | | | | | | | | | | | | |
| RDW | −0.18 | −0.10 | −0.11 | 0.97 ** | 1 | | | | | | | | | | | | |
| SFW | 0.75 ** | 0.83 ** | 0.85 ** | 0.10 | 0.12 | 1 | | | | | | | | | | | |
| SDW | 0.74 ** | 0.81 ** | 0.86 ** | 0.08 | 0.11 | 0.97 ** | 1 | | | | | | | | | | |
| LFW | 0.99 ** | 0.98 ** | 0.94 ** | −0.17 | −0.14 | 0.73 ** | 0.70 ** | 1 | | | | | | | | | |
| LDW | 0.98 ** | 0.96 ** | 0.93 ** | −0.18 | −0.14 | 0.71 ** | 0.67 ** | 0.98 ** | 1 | | | | | | | | |
| FFW | 0.67 ** | 0.72 ** | 0.77 ** | 0.00 | 0.01 | 0.65 ** | 0.71 ** | 0.60 ** | 0.58 ** | 1 | | | | | | | |
| FDW | 0.72 ** | 0.76 ** | 0.79 ** | −0.05 | −0.05 | 0.61 ** | 0.67 ** | 0.67 ** | 0.64 ** | 0.97 ** | 1 | | | | | | |
| SW | −0.11 | −0.10 | −0.17 | −0.31 | −0.20 | 0.17 | 0.26 | 0.01 | −0.10 | −0.53 ** | −0.47 ** | 1 | | | | | |
| IC50 | −0.25 | −0.30 | −0.23 | 0.00 | 0.00 | −0.30 | −0.16 | 0.35 * | −0.35 * | 0.14 | 0.10 | −0.52 ** | 1 | | | | |
| TP | 0.02 | −0.01 | 0.02 | 0.10 | 0.05 | 0.03 | 0.13 | −0.03 | −0.03 | 0.06 | −0.01 | −0.44 ** | 0.42 * | 1 | | | |
| TF | 0.12 | 0.08 | 0.12 | −0.07 | 0.00 | 0.12 | 0.24 | 0.03 | 0.05 | 0.31 | 0.20 | −0.28 | 0.56 ** | 0.37 * | 1 | | |
| RAC | 0. 30 | 0.37 * | 0.29 | 0.03 | 0.10 | 0.29 | 0.18 | 0.41 * | 0.35 * | 0.00 | 0.08 | 0.48 * | −0.47 * | −0.28 | −0.37 * | 1 | |
| EOC | 0.25 | 0.26 | 0.29 | −0.30 | −0.20 | 0.18 | 0.20 | 0.25 | 0.22 | 0.26 | 0.23 | 0.24 | 0.05 | −0.26 | 0.49 ** | 0.22 | 1 |

[1] LA, Leaf area; TFW, Total fresh weight; TDW, Total dry weight; RFW, Root fresh weight; RDW, Root dry weight; SFW, Stem fresh weight; SDW, Stem dry weight; LFW, Leaf fresh weight; LDW, Leaf dry weight; FFW, Flower fresh weight; FDW, Flower dry weight; SW, 1000 Seeds weight; IC50, Antioxidant activity; TP, Total phenolic compounds; TF, Total flavonoids; RAC, Rosmarinic acid content; EOC, Essential oil content. ** $p \leq 0.01$, * $p \leq 0.05$.

### 4. Conclusions

The present study illustrated that cultivated *Salvia* species are valuable sources of antioxidant and flavonoid compounds. The cultivation of *Salvia* led to an improvement in the quantity and quality of medicinal compounds; it is also possible to improve production management. Growth parameters, such as the ratio of leaf dry weight/total dry weight, fresh dry weight/total dry weight, and root architecture, were closely correlated with the geographic origins of the species. The biochemical content was strongly dependent on the species and plant organs.

The GC-MS coupled with principal component analysis of various *Salvia* species revealed diverse compound distributions. *Salvia virgata*, *S. syriaca*, and *S. officinalis* were separated from the other species. β-caryophyllene, caryophyllene oxide, and germacrene D were the main compounds in the *Salvia* species.

In this study, environmental conditions and agronomic practices were constant for all species; therefore, the variations in the recorded factors were attributed to genetic differences. Overall, *Salvia* species have the greatest potential to profit from the production of medicinal compounds, such as phenol and flavonoid compounds and essential oils. Cultivation and production of these species not only lead to the achievement of significant amounts of these compounds but also provide the possibility of managing the production and preservation of plants.

The present study is a small step toward introducing the potential of this genus and the possibility of domestication of some species. Cultivation of medicinal plants has become the preservation of the biodiversity of the country's species and prevents the destruction of pastures and the extinction of wild species as a result of incorrect and unstable harvests. The domestication and cultivation of these species of agriculture lead to the production of uniform raw materials, the possibility of production in all seasons (according to the climatic diversity of the country), the management of production in line with national and international markets, and mechanization. Moreover, we can also benefit from the planting, harvesting, and processing stages and the possibility of breeding and producing new cultivars. It is hoped that future research in the next stages on topics such as the modification and purification of compounds will be effective in the production of pharmaceutical products.

**Supplementary Materials:** The following supporting information can be downloaded at: https://www.mdpi.com/article/10.3390/agronomy12102455/s1, Figure S1. Calibration curve of standard gallic acid for determination of total phenolic. Figure S2. Calibration curve of standard quercetin for determination of flavonoid. Figure S3. The ratio of the dry weight of various organs to total dry weight in Salvia species. TDW, Total dry weight; RDW, Root dry weight; SDW, Stem dry weight; LDW, Leaf dry weight; FDW, Flower dry weight. Table S1. Plant materials and collection details of studied Salvia species. Table S2. Climatic parameters of the experimental field averaged over during the growth period.

**Author Contributions:** G.E.: Investigation and methodology; H.F.: formal analysis, review, and editing; M.B.a.: writing—review and editing; M.A.: supervision, conceptualization, data curation, manuscript editing and finalizing, funding acquisition; H.A.: advisor, research design advising; J.V.: advisor, research design advising. Y.F.: Project administration, supervision, conceptualization, data curation, funding acquisition. All authors have read and agreed to the published version of the manuscript.

**Funding:** This research was supported by Ferdowsi University of Mashhad under research grant No. 41387. This study was supported partially by a grant from JST CREST (Grant Number JPMJCR17 O2), Japan.

**Institutional Review Board Statement:** Not applicable.

**Informed Consent Statement:** Not applicable.

**Data Availability Statement:** The data presented in this study are available upon request from the corresponding author.

**Acknowledgments:** The authors appreciate Mohammad Reza Joharchi, a botanist from the Research Center from Plant Science, for the identification of plant species.

**Conflicts of Interest:** The authors declare no conflict of interest.

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
