# Peer review of "Diversity of Chemical Composition and Morphological Traits of Eight Iranian Wild Salvia Species during the First Step of Domestication"

_agronomy, doi:10.3390/agronomy12102455_

Round 1
Reviewer 1 Report
Dear authors, I appreciate your work.The research is conducted correctly in terms of the sequence of steps.
Generally, actual results are clearly presented, the discussions are relevant and support the conclusions. The bibliography is current and exhaustive, correctly written.
However, additional date and major corrections are required. 1. Use italic font throughout the text for ,,Salvia” , ,,Lamiaceae” in all text (abstract and full text) 2. Line 42: write ,,use plants for treatment” or ,,use medicinal plants” 3. Line 61: ,,allelopathic” instead of ,,Allelopathic”4. At point 2.5.: give more details (the volume / amount of methanol used, the amount of extract obtained….).
5. Point 2.5.1. replace ,,phenols” with ,,phenolic compounds” (it is only a recommendation / suggestion). Make this change in whole text.
6. Point 2.5.1, 2..5.2., 2.5.3.: use the full names of reagents, not formulars ( ,,sodium carbonate” instead of ,,Na2CO3 “ …..)
7. Upload complete data about gallic acid calibration curve and quercetin calibration curve in Supplementary Materials
8. In all formulas and corresponding text replace ,,absorption” with ,,absorbance”
9. Point 2.5.4.: your method isn’t selective for rosmarinic acid in herbal solutions. Can you replace it with another, more selective?
10. Point 2.5.4.: In formulas 3: explain ,,34105.2”
11. Point 2.5.4.: match the legends of formulas 2 and 3 (A is identical to A333)
12. Line 166: ,,3” instead ,,three”
13. Line 177: ,,100” instead ,,one hundread”
14. Line 178: the text is incomplete - ,,for four …?”
15. Line 197: ,,indices to” instead ,,indicesto”
16. Point 2.5.6.: justify why you used two GC methods
17. Point 3: The result and the discussions should be separately presented; if not, I think the title for Point 3 must be ,,Results and discussions”
18. Put all the results (in figures and tables) in the same sequence of samples (for example, the sequence used in table 1: S. atropanata, S. macrosiphon, ….)
19. Some comments regarding the relevance of this study are necessary at point 3, taking into account that methanol extracts are phytochemical and biochemical analysed, methanol is a toxic solvent (class 2) and the final goal is pharmaceutical valorization.
20. Line 276: replace ,,tissue” with ,,samples”
21. Line 277: ,,*” ?
22. Lines 282-284: revise the text
23. Line 286: ,, Thoma et al. “ instead of ,, Thoma et al”
24. Lines 288-289: revise the text (,,Phenolic compounds….acting as ? electrons”)
25. Lines 290, 307, 316, 318, table 2 s.a. : ,,total phenols” instead ,,total phenol” (If you keep ,,phenols”
26. at point 2.5.1.)
27. Line 305: include a short explanation about the role of enzyme Phenylalanine ammonia-lyase.
28. Line 307: replace ,,flavone & flavonols, and flavonoid content” with ,,total flavonoids content”
29. Lines 320-322: revise the text (,, …followed by...” ?); carnasol, rosmarinic acid, carnosic acid, caffeic acid, rosmanol, and rosmadial are phenolic compounds, gengwanin and cirsimaritin are flavonoids
30. Line 326: write ,, 121.7 mg QE/g extract, and 156.7 mg QE/g extract”
31. Line 330, 331: write ,, root: 71.7 QE/g extract, leaf: 61.5 QE/g extract”
32. Line 332: write ,,per ha” instead ,,per h”
33. Line 341 – 342: move the text to phenolic compounds
34. Line 379: delete the abbreviation / explanation for HPLC
35. Line 381-382: complete the text (S. brachyantha… vegetal organ and solvent used), and clarify the text (,,rosmarinic acid has been introduced ??? as the main compound”)
36. Line 428: ,,Salvia” instead ,,salvia”
In conclusion, I consider that the manuscript can be published only after major revis.
Author Response
Reviewer 1
Dear authors, I appreciate your work.
The research is conducted correctly in terms of the sequence of steps.
Generally, actual results are clearly presented, the discussions are relevant and support the conclusions. The bibliography is current and exhaustive, correctly written.
However, additional date and major corrections are required.
Dear Editor and reviewers
We thank the reviewer for their careful reading of the manuscript and their constructive remarks. We have taken the comments on board to improve and clarify the manuscript. Please find below a detailed point-by-point response to all comments (reviewers’ comments in black, our replies in blue).
Best Regards
Majid Azizi
Yashiharu Fujii
- Use italic font throughout the text for ,,Salvia” Lamiaceae” in all text (abstract and full text)
Ans. We corrected in whole text..
- Line 42: write ,,use plants for treatment” or ,,use medicinal plants”
Ans: We write “use medicinal plant”.
- Line 61: ,,allelopathic” instead of ,,Allelopathic”
Ans: Its changed.
- At point 2.5.: give more details (the volume / amount of methanol used, the amount of extract obtained….).
Ans: It was added.
- Point 2.5.1. replace ,,phenols” with ,,phenolic compounds” (it is only a recommendation / suggestion). Make this change in whole text.
Ans: We replaced all.
- Point 2.5.1, 2..5.2., 2.5.3.: use the full names of reagents, not formulars ( ,,sodium carbonate” instead of ,,Na2CO3 “ …..)
Ans: Its true, we changed all.
- Upload complete data about gallic acid calibration curve and quercetin calibration curve in Supplementary Materials
Ans: we added in supplementary.
- In all formulas and corresponding text replace ,,absorption” with ,,absorbance”
Ans: We changed all.
- Point 2.5.4.: your method isn’t selective for rosmarinic acid in herbal solutions. Can you replace it with another, more selective?
Ans: Thanks for the precise suggestion. But we used the method was explained in ref. 34 and 35 and we know that both are old and it was better to do using HPLC. For screening the spectrophotometric may be very fast and Accessible. O also informed you that but this work is finished and we dont have more samples for repeating this parameter.
- Point 2.5.4.: In formulas 3: explain ,,34105.2”.
Ans: 34105.2 is a dilution factor (Komali and Shetty, 1998). It was added to line 167.
- Point 2.5.4.: match the legends of formulas 2 and 3 (A is identical to A333)
Ans: Formula 2 have used to estimate Rosmarinic acid content in fresh tissue and formula 3, expressed RAC on the basis of plant dry weight. The moisture content is determined according to the following formula,
%Mw =(Ww-Wd /Ww) * 100
Where, Mw is moisture content in percent and Ww, Wd, are fresh weight (FW) and dry weight (DW) respectively.
- Line 166: ,,3” instead ,,three”
Ans: It replaced.
- Line 177: ,,100” instead ,,one hundread”
Ans: It changed.
- Line 178: the text is incomplete - ,,for four …?”
Ans: It was completed.
- Line 197: ,,indices to” instead ,,indicesto”
Ans: It was completed.
- Point 2.5.6.: justify why you used two GC methods
Ans: as you know GC uses to separate the different molecule types of essential oils according to their weight, size and their affinity to the column used. So GC do not detect the exact compounds. Also, in GC analysis if the molecules are close in weight and affinity the separating of compounds are not clear. first, we used a GC-MS to detect compounds from its library in essential oil. For other samples we only used GC. The GC separates the compounds from each other and gives the RI, while the mass spectrometer helps to identify them based on their fragmentation pattern.
- Point 3: The result and the discussions should be separately presented; if not, I think the title for Point 3 must be ,,Results and discussions”
Ans: That’s true we corrected Point 3 as “Results and discussions”
- Put all the results (in figures and tables) in the same sequence of samples (for example, the sequence used in table 1: S. atropanata, S. macrosiphon, ….)
Ans: All the Figures and Table corrected in the same sequence of samples..
- Some comments regarding the relevance of this study are necessary at point 3, taking into account that methanol extracts are phytochemical and biochemical analysed, methanol is a toxic solvent (class 2) and the final goal is pharmaceutical valorization.
Ans: That’s true final goal is pharmaceutical valorization but we just measure some parameters such as phenol, flavonoid, and … with MeOH extraction. Because it’s one of the common methods for measurement of these parameters.
- Line 276: replace ,,tissue” with ,,samples”
Ans: It was replaced.
- Line 277: ,,*” ?
Ans: It was replaced.
- Lines 282-284: revise the text
Ans: It was replaced.
- Line 286: ,, Thoma et al. “ instead of ,, Thoma et al”
Ans: It was replaced. Endnote software automatically corrected it.
.
- Lines 288-289: revise the text (,,Phenolic compounds….acting as ? electrons”)
Ans: It was replaced.
- Lines 290, 307, 316, 318, table 2 s.a. : ,,total phenols” instead ,,total phenol” (If you keep ,,phenols” 26. at point 2.5.1.)
Ans: We replaced phenols with phenolic compounds.
- Line 305: include a short explanation about the role of enzyme Phenylalanine ammonia-lyase.
Ans: A short explanation about the role of the enzyme Phenylalanine ammonia-lyase was added.
- Line 307: replace ,,flavone & flavonols, and flavonoid content” with ,,total flavonoids content”
Ans: It was replaced.
- Lines 320-322: revise the text (,, …followed by...” ?); carnasol, rosmarinic acid, carnosic acid, caffeic acid, rosmanol, and rosmadial are phenolic compounds, gengwanin and cirsimaritin are flavonoids
Ans: It was corrected as :
The phenolic compounds are isolated from the extract of the Salvia officinalis L., such as carnosol, rosmarinic acid, and carnosic acid, followed by caffeic acid, rosmanol, rosmadial, genkwanin, and cirsimaritin as flavonoid compound which has the most effective antioxidant activity [53, 54]. Koşar, Göger [31]Koşar et al. (2008) reported TP (28.3-212.3 GAE/g extract) in various extracts of S. virgata aerial parts.
- Line 326: write ,, 121.7 mg QE/g extract, and 156.7 mg QE/g extract”
Ans: It was corrected.
- Line 330, 331: write ,, root: 71.7 QE/g extract, leaf: 61.5 QE/g extract”
Ans: It was corrected.
- Line 332: write ,,per ha” instead ,,per h”
Ans: It was corrected.
- Line 341 – 342: move the text to phenolic compounds
Ans: It was corrected.
- Line 379: delete the abbreviation / explanation for HPLC
Ans: It was removed.
- Line 381-382: complete the text (S. brachyantha… vegetal organ and solvent used), and clarify the text (,,rosmarinic acid has been introduced ??? as the main compound”)
Ans: It was corrected.
- Line 428: ,,Salvia” instead ,,salvia”
Ans: It was corrected.
In conclusion, I consider that the manuscript can be published only after major revis.

Reviewer 2 Report
The article discusses the 11 Salvia genotypes in great detail. The results are presented in an interesting and very diverse way. However, I have two fundamental points. The chapter Conclusions is more like an introduction. The authors compared 11 different plants - there is no summary that is best suited for cultivation. Is it possible, on the basis of the results obtained by the authors, to select species for the further process of introducing to cultivation?
The article also lacks an explanation as to why these species were used in the experiment. What the authors based their choice on.
More detailed comments in the text.

Author Response
Reviewer 2
The article discusses the 11 Salvia genotypes in great detail. The results are presented in an interesting and very diverse way. However, I have two fundamental points. The chapter Conclusions is more like an introduction. The authors compared 11 different plants - there is no summary that is best suited for cultivation. Is it possible, on the basis of the results obtained by the authors, to select species for the further process of introducing to cultivation?
Ans: Dear Editor and reviewers
We thank the reviewer for their careful reading of the manuscript and their constructive remarks. We have taken the comments on board to improve and clarify the manuscript. Please find below a detailed point-by-point response to all comments (reviewers’ comments in black, our replies in blue).
Best Regards
Majid Azizi
Yashiharu Fujii
The article also lacks an explanation as to why these species were used in the experiment. What the authors based their choice on.
Ans; Most of the species were collected from the center (Isfahan province) and North-east (Khorasan province) of Iran, where these plants started to be domesticated and cultivated. There are more than 60 Salvia species in Iran. In this study, we chose S. officinalis as the main Salvia species that use as herbs in various cultures and drugs and S. sclarea as another important species in the new pharmacy industry. Other species are selected according to their importance in Iran and some try to cultivate them in Iran.
More detailed comments in the text.
17àSalvia as Latin name of the genus you should write italicized, please change in the whole article, you can also can use the English name 'sage'All salvia in text change to italic
Ans: All “Salvia” genus corrected to Italic format.
36àthe same word is in a title
Ans: Domestication removed from Keywords
Line 44: This is too general a statement in most European countries that sage is cultivated.
Ans: It was corrected as : Although in most European countries Salvia is cultivated in the field but some medicinal plants are being harvested from nature, not only for traditional medicinal purposes but also for trade and commerce
Line 60: Allelopathic activity corrected as: “ allelopathic activity”
127à Why harvested in 50% of plant blooming?
Ans: Generally, the highest biochemical constitute (especially essential oil content) in Lamiaceae species are in blooming stage (Hornok, L. Cultivation and processing of medicinal plants)
131àin this temperature we can achieve only air dry weight not total (in this case herbal raw material should be dried at temperature 105 °C for 24 h or at 70 °C for 72 h.)
We dried sample in 35°C because one of our aim was measuring phytochemical parameters. As we know the best drying temperature to save the most essential oil content in herbs is about 30-40°C. (Hornok, L. Cultivation and processing of medicinal plants). The temperature also uses in industry and herbs drugs companies. In the other part of the research, we used 105°C to obtain the moisture content in various plants parts (data not shown).
179 à corrected
202àIf I understand correctly, 9 plants were finally measured, this is a bit too little to get reasonably accurate results
Ans: We had 3 replication for each treatment and 12 plant in each replication so we have 36 data for each treatment (each Salvia species).
220 S. macrosiphonà S. macrosiphon
225 Where are the letters. In my opinion they should been put into table to better analyse the results. Additional it would be better to divide this big tabel to two smaller. However, in my opinion, standard deviations can be ignored. Then the tables will become clearer.
Ans: The Table was corrected but because of many tables and figures we decided don’t divide this Table.
227à corrected
302 Echinacea purpureaàEchinacea purpurea
307 Its corrected.
448 S. virgatawasàS. virgata was
462 (Table5) à (Table 5)
Please put the space between numbers and letters.
It was corrected.

Round 2
Reviewer 1 Report
Dear Authors,
The new version of the manuscript responds to all my previous requests.
In conclusion, I consider that the manuscript can be published in this revised version.
I have only a minor remark: at line 42-43: delete ,,for treatment”; it is a pleonasm in the text.
Congratulations!
Author Response
Cover letter
Dear Editor, Agronomy
We would like to thank you for giving the authors the opportunity to submit a revised version of the manuscript titled “Diversity of Chemical Composition and Morphological Traits of Eight Iranian Wild Salvia Species during the First Step of Domestication” to agronomy. We would also like to express our appreciation to the reviewers for the time they have taken to review this manuscript. We believe the comments and suggestions of the reviewers have helped improve the content of this manuscript.
I removed the "…for treatment” at line 42-43.
Sincerely yours,
Majid Azizi ([email protected])
Yoshiharu Fujii ([email protected])
